# How Replicates Can Inform Potential Users of a Measurement Procedure about Measurement Error: Basic Concepts and Methods

**DOI:** 10.3390/diagnostics11020162

**Published:** 2021-01-22

**Authors:** Werner Vach, Oke Gerke

**Affiliations:** 1Basel Academy for Quality and Research in Medicine, 4051 Basel, Switzerland; 2Integrative Prehistory and Archeological Science (IPAS), Department of Environmental Sciences, University of Basel, 4055 Basel, Switzerland; 3Department of Nuclear Medicine, Odense University Hospital, 5000 Odense, Denmark; oke.gerke@rsyd.dk; 4Department of Clinical Research, University of Southern Denmark, 5230 Odense, Denmark

**Keywords:** measurement error, replicates, agreement, repeatability, reproducibility, reliability

## Abstract

Measurement procedures are not error-free. Potential users of a measurement procedure need to know the expected magnitude of the measurement error in order to justify its use, in particular in health care settings. Gold standard procedures providing exact measurements for comparisons are often lacking. Consequently, scientific investigations of the measurement error are often based on using replicates. However, a standardized terminology (and partially also methodology) for such investigations is lacking. In this paper, we explain the basic conceptual approach of such investigations with minimal reference to existing terminology and describe the link to the existing general statistical methodology. This way, some of the key measures used in such investigations can be explained in a simple manner and some light can be shed on existing terminology. We encourage clearly conceptually distinguishing between investigations of the measurement error of a single measurement procedure and the comparison between different measurement procedures or observers. We also identify an unused potential for more advanced statistical analyses in scientific investigations of the measurement error.

## 1. Introduction

When a subject is contacting the health care system, it expects optimal advice or treatment. Healthcare providers can provide this only if they have a clear picture about the status of the current physical, physiological, psychological, mental and social status of the subject. Hence health care providers apply a wide range of measurement procedures to obtain numbers containing corresponding information—and to an increasing degree, subjects use such procedures to assess their state by themselves. These procedures can range from single questions directly addressed to the subject over questionnaires, structured interviews, simple measurements of physical and physiological parameters (such as blood pressure, body temperature, or height and weight) to advanced methods, such as laboratory tests, imaging procedures, or automated decision tools based on artificial intelligence. All these measurement procedures have one feature in common: they are not error-free.

How can health care providers (or other users) trust the use of a measurement procedure, if this procedure is not error-free? In an ideal world, this requires one to ensure that the measurement error is of negligible magnitude or at least to inform the potential users of a procedure about the magnitude, such that they can take this knowledge into account in the subsequent decision-making process. In any case, this requires knowledge about the magnitude of the measurement error.

This knowledge can be generated by scientific investigations performing a systematic assessment of the measurement error of a procedure. Whereas the measurement error of a single measurement is a unique, single number (the difference between the measured value and the true value), the measurement error of a measurement procedure is a (statistical) distribution, describing the distribution of the individual measurement errors when applying the procedure in many subjects. Scientific investigations of the measurement error are hence nearly always based on applying the procedure of interest in a relevant sample of subjects and to study the distribution of the individual measurement errors. There are two different approaches to determine the individual measurement errors in such studies: if a gold standard measurement procedure is available, i.e., a method allowing us to measure the true value of the quantity of interest in an error-free manner, the individual measurement error can be observed directly. If this is not the case, we can apply the measurement procedure repeatedly and try to approximate the true value from the replicates.

In the context of such investigations, a variety of conceptual terms, such as reproducibility, reliability, agreement, accuracy or precision, are in common use, although their use is often inconsistent [1,2]. This has raised some confusion, which is exaggerated by a variety of statistical approaches and methods used in these investigations [3,4,5,6] which makes it particularly challenging to develop reporting guidelines [7]. Consequently, in this paper we try to explain the basic rationale (and the link to standard statistical techniques) used in this type of investigation with minimal reference to such terms. We take the simple starting point that the aim of such investigations is to inform potential users of a measurement procedure about the measurement error of this procedure, and that it is rather clear what we mean with this if a gold standard method to measure the true value is available. A step-by-step approach is chosen to depict the way from a simple study using a gold standard measurement procedure to studies using replicates, as well as potential strategies for analyzing such studies and informing potential users. The difference between measurement error and repetition error will play a central role. Each step is devoted to a separate issue, but also the relation between different issues is described.

## 2. Choice of Populations and Observers

Irrespective of using a gold standard measurement procedure or using replicates, a sample of subjects to which the procedure is applied has to be studied, because statements about the negligible magnitude of the measurement error are not statements about the actual discrepancy between the measured value and the true one in a single subject, but about the magnitude we can expect in a single subject based on the knowledge about the distribution we can observe in many subjects, similar to the subject of interest. Hence the generation of such a sample is one basic step in any scientific investigation on the measurement error.

Generating such a sample can be a challenging and crucial task, as the measurement error of a procedure may differ between different groups of subjects. For example, an accelerometer may give a rather realistic pattern about the daily steps of a subject, if the subject is young and moving with high energy, but it might fail in an old subject moving around slowly after hip surgery. It is hence essential to choose (preferably random) samples from populations that are relevant to inform potential users. Often, these will be populations approaching the health care system in a specific situation, in which a specific procedure is typically used. For example, parents of a newborn child may be interested to understand the measurement error of a bilirubin assessment performed on their child to make a decision about phototherapy. They are not interested in knowing the measurement error of the same procedure when used to detect liver damage in adults. Similarly it makes little sense to inform a geriatrician about the measurement error of an accelerometer based on a study in adolescents.

However, choosing subjects is only one aspect. The measurement error of a procedure is also often depending on the subject performing the procedure, reading of the result(s) or interpreting them. In the following such a person is called an “observer”—a term widely used in the literature, even if the role of this subject is not necessarily to make an observation. Observers (often also called “raters”) may vary in experience, education, training or profession, and this can have a substantial impact on the measurement error of the procedure they use. The measurement error may also depend on observer characteristics, such as gender or body proportions: some measurement procedures may be easier to perform if you have thin fingers. Since the potential user is usually lacking a precise knowledge about the relevant personal characteristics of the observer involved in the procedure, it is essential to inform the user with respect to a relevant population of observers. For example, if radiological images are interpreted in a hospital by lab assistants, it is not useful to inform patients about the measurement error based on a study using senior physicians as observers.

So, in general, a good scientific investigation of the measurement error is characterized by randomly selecting both subjects from a relevant subject population and observers from a relevant population of observers. In addition, observers should be matched to subjects in a random or controlled manner as there may be subtle interactions between observers and subjects. For example, some observers may be well-suited to interrogating subjects with low level of education, but fail in subjects with high education, and in others it may be the other way round. In some instances, observers may not even be human beings, but technical devices.

## 3. Informing about the Magnitude of a Measurement Error If a Gold Standard Measurement Procedure Is Available

If a gold standard measurement procedure is available, in a relevant sample both the measurement procedure and the gold standard method can be applied. We denote in the following the value obtained by the measurement procedure with subject *i* with $y_{i}$ and the value obtained by the gold standard method with $y_{i}^{*}$, i.e., the latter is a gold standard measurement and hence identical to the true value of the quantity of interest. In this case, the individual measurement error
$$e_{i}=y_{i}-y_{i}^{*}$$
can directly be observed in each subject. Consequently, we can directly analyze the (theoretical) distribution of the measurement error based on the empirical distribution observed in our sample. For example, we can present the empirical distribution by a histogram and can mark certain characteristics of this distribution, as shown for an artificial example in Figure 1. In describing this distribution, two aspects can be distinguished. First, we can look at location measures, such as the mean, the median or the mode of the distribution. These numbers inform us about the systematic bias of the measurement procedure, i.e., whether there is a tendency of the method to produce “on average” values above or below the true value. Second, we can investigate the spread of the distribution, informing us about how far single measurements can deviate from the true value. One very useful approach is to report the upper and lower 2.5% percentile, as a range covering 95% of all individual measurement errors is obtained in this way. The interpretation of such an interval is very similar to that of normal intervals or normal ranges, which many health care providers are familiar with.

However, to obtain precise estimates of these two percentiles, rather large sample sizes are required [8,9]. Hence often the computation of the percentiles is based on the assumption of a normal distribution, allowing us to obtain more precise estimates of the percentiles based on the estimated mean $\hat{\mu }$ and the estimated standard deviation $\hat{\sigma }$ using the famous formula $\hat{\mu }\pm 1.96\hat{\sigma }$. A histogram (or better a normal probability plot) can be helpful to judge the assumption of normality.

It should be emphasized that the essential property of the distribution of the measurement error is the spread, and not the (systematic) bias. If a measurement error is biased, we can estimate the bias based on a sufficiently large sample and then correct for this bias by subtracting it from the single measurements. However, there is no way to correct for the spread, even if the percentiles or the standard deviation are known.

## 4. Judging the Relevance of the Magnitude of a Measurement Error

As mentioned in the introduction, the aim of scientific investigations on the measurement error is to inform potential users whether the measurement error is of negligible or at least acceptable magnitude. The numbers mentioned in the previous section do not provide this directly. They have to be checked for their (clinical) relevance. One way would be to compare them with predefined relevance limits.

However, even if the specification of such limits is widely recommended (cf. [10]), such relevance limits are rarely specified in scientific investigations of the measurement error. Typically, it is assumed that the users can judge the numerical information about the magnitude of the measurement error based on their familiarity with the quantity of interest in their use in routine practice. For example, it is assumed that a physician can relate a 95% range of [−0.25, 0.37] for the measurement error of a procedure to assess a patient’s body temperature (in °C) to his or her daily work, as the physician has an idea about how often a measurement error of this magnitude will lead to a difference in the management of the patient.

If the influence of a measured quantity on the management can be reduced to a classification based on exact thresholds (e.g., the WHO-classification of the body mass index in Europe, https://www.euro.who.int/en/health-topics/disease-prevention/nutrition/a-healthy-lifestyle/body-mass-index-bmi), it might be argued that such thresholds (or more precisely the difference between the thresholds) can be used to judge the magnitude of the measurement error. However, in such a case, a user will probably be more interested in the measurement error of the derived classification.

A special case appears if a measurement procedure is used repeatedly in a subject to assess a change over time, in particular, after an intervention. For many quantities, there have been attempts to assess a so-called minimal important difference (MID), i.e., a value aiming to describe a change that is relevant for subjects in some sense. Then it can be argued that the measurement error should be small enough to allow with high probability to detect a change at least in the magnitude of the MID. However, this special case is also special with respect to other aspects. We will come back to this later in Section 16.1.

## 5. Replacing a Gold Standard Measurement by Replicates

If a gold standard measurement procedure to measure the true value is not available, it is a rather straightforward idea to repeat the procedure of interest several times and to take the average value of these repeated measurements in order to obtain an approximation of the true value. There are two basic approaches to generate such replicates. The first one is the repetition in time, i.e., the same observers repeat the whole measurement procedure after some time. The second one is the repetition by different observers, which is often the more attractive approach, in particular if several observers are already involved in the study. Using different observers makes it easier to ensure some degree of independence between the replicates, as the observers can be blinded for each other. If the same observers are used over time, there is some danger that they can remember the results of the previous application of the procedure in a subject.

Taking the average over such replicates is a very well known scientific procedure to reduce the magnitude of an individual measurement error (e.g., in experimental research), as the stochastic uncertainty of an average is always smaller than that of the single measurement. However, in our context this is not sufficient, which is explained in the sequel. Let $y_{ir}$ denote the measurement of replicate *r* in subject *i*. The basic idea is to use the average ${\bar{y}}_{i.}$ over the replicates in subject *i* as a substitute for the true value $y_{i}^{*}$. This requires that this average gets close to the true value $y_{i}^{*}$, i.e., there are many replicates such that the stochastic uncertainty becomes negligible. Formally, we can introduce the symbol $\mu_{i}$ to denote the expected value of $y_{ir}$ (which can be also imagined as the value we should obtain when averaging over an infinite number of replicates). So, the basic assumption made when using replicates is simply
$$\mu_{i}=y_{i}^{*}.$$

When can this property be expected to hold? It requires that the replicates do not replicate (in part) the error. Let us look at a simple example. In cancer patients, it is rather common to use PET/CT for measuring the size of the active part of a tumor—for example, after radiation therapy. This requires that the observer marks the region of interest and a machine (e.g., a scanner) can then compute the size of the active region. If the tumor is now surrounded by an inflammation, this looks very similar to the active tumor, and neither the observer nor the machine can distinguish this, and the tumor size is overestimated to a substantial degree. Additionally, this part of the error will be made by any replicate, even if it is performed by another observer.

Statistically speaking, the essential property of replicates is that they are uncorrelated, given the true value. This means that the differences between the measured values of the replicates and the true value, i.e., the measurement errors
$$e_{ir}=y_{ir}-y_{i}^{*}$$
of the different repetitions are uncorrelated within an individual. In particular the direction of the error of the first replicate may not inform us about the direction of the error of the subsequent replicates. Actually, there are many situations where this might be the case, in particular if the error is related to characteristics of the subject that do not change over the replicates. If a measurement procedure overestimates the true value in female, old or highly educated subjects and underestimates it in male, young or less educated subjects, respectively, such issues are never detected in a study based on replicates. Additionally, many of these issues cannot be avoided by a careful design of the replication procedure, such as blinding. This applies only to correlations between replicates due to remembering or knowing other replicates.

In the following sections, we use the term “perfect replicates” if they are uncorrelated, given the true value. If replicates are not perfect, they are typically positively correlated, and this implies that the variation in the true measurement error distribution is underestimated (when the methods outlined in the subsequent sections are used). So, in general, the measurement error studies, based on replicates, always pose a risk of underestimating the measurement error of a measurement procedure, as replicates are often imperfect. This is a property we should always have in mind when interpreting the results of a measurement error study based on replicates.

## 6. A Short Reflection about Perfect Replicates and the Absence of a Truth

Taking the difficulties to generate perfect replicates into consideration, it might be surprising that there are so many studies using replicates and so few using gold standard measurements. A simple reason is the lack of gold standard methods. However, this often reflects a somewhat deeper problem: it is often difficult or impossible to define or even imagine the “true value” (at least as a specific number). This is very obvious for most psychometric instruments trying to measure constructs, such as quality of life or pain intensity. However, even for many physiological parameters, such as body temperature or blood pressure, it is not easy to define a true value due to biological variation within the body or within rather short time periods. Additionally, a quantity such as “tumor size” is more a construct than a true value, as it would require exactly defining the border surface of the tumor.

Consequently, in many fields, the only practical way to “define” the true value may be to regard the average over (perfect) replicates as the true value. Such a pragmatic approach can be also found explicitly in the literature, as discussed in [2]. However, we do not follow this approach here, as it has a conceptual drawback. The values $\mu_{i}$ do not only depend on the measurement procedure and the population of observers, but also on the way we generate the replicates. For example, if we fail to blind observers from each other, there is a risk that all subsequent observers try to reproduce the values of the first observer. This means that $\mu_{i}$ tends to get close to $y_{i1}$, i.e., the value produced by the first observer. If the observers are blinded, $\mu_{i}$ will be different and not particular close to the value of one observer. This type of “mistake” cannot be described if we just regard $\mu_{i}$ as the true values. Or, in other words, we can have different true values depending on how the replicates are generated.

Consequently, we avoid equating $\mu_{i}$ with the true value $y_{i}^{*}$. Instead, we explicitly consider the (unobservable) quantities
$$\epsilon_{ir}=y_{ir}-\mu_{i},$$
i.e., the “measurement error” of each replicate. We refer to the latter as the individual repetition error and to its distribution as the repetition error (assuming for the moment that all subjects have the same distribution). However, we have just seen that there can be different $\mu_{i}$ depending on how the replicates are generated; hence, we should keep in mind that the repetition error is a property of the measurement procedure and the way to generate replicates. In particular, generating replicates over time results in a different repetition error than using different observers. This distinction is, for example, implied by terms such as intra- and inter-observer variability. This first term refers to replicates over time within one observer, and the latter to replicates using different observers. Test–retest variability also refers to a specific way to generate replicates in time, namely to re-administer a procedure after a specific time. However, all these terms reflect only the intention of generating replicates, but do not define a unique repetition error distribution. It still depends on the details of how to generate replicates, e.g., on the degree blinding could be ensured or on the distance in time chosen.

## 7. Estimating Characteristics of the Repetition Error Distribution Based on Replicates

In order to get insights into the repetition error distribution, the differences $\delta_{ir}=y_{ir}-{\bar{y}}_{i.}$ should not be used directly as approximation of the individual repetition error, as they tend to have within each subject a lower spread than the values $\epsilon_{ir}$. This is a well known phenomenon due to using the observations both with respect to estimating the true mean $\mu_{i}$ as well as to describing the deviation from the true mean. (The most prominent consequence is the fact that, in estimating a population standard deviation from a sample of size *n*, we divide by n−1 instead of *n* in order to obtain an unbiased estimate).

A standard approach to solve this problem is to use a simple repetition error model, which reads
(1)$$y_{ir}=\mu_{i}+\epsilon_{ir}\;\mathrm{with}\;r=1,\ldots ,R\geq 2,E\;\epsilon_{ir}=0\;\mathrm{and}\;\mathrm{Var}\;\epsilon_{ir}={\sigma_{\epsilon }}^{2}\;$$
such that $\sigma_{\epsilon }$ denotes the standard deviation of the repetition error. The standard statistical technique to fit the model (1) is the ordinary least squares principle. Statistical software for both one-way ANOVA and regression (with the subject as categorical covariate) can be used. The estimate for $\sigma_{\epsilon }$ is typically reported as the RMSE (root mean squared error).

## 8. Improved Informing by Translating the Standard Deviation

There are mathematical and historical reasons why standard statistical techniques to fit the simple repetition error model (1) provide an estimate of the standard deviation of the measurement error distribution and not of other characteristics of interest. In any case, the standard deviation was not among the parameters we mentioned above in describing the measurement error distribution; it only appeared as an auxiliary quantity to compute a 95% range. This is due to the simple fact that standard deviations themselves have no directly appealing interpretation, at least if compared with other quantities, such as the 95% range.

Indeed, there are different approaches to transform the standard deviation of the repetition error into numbers that are easier to interpret for a potential user of the measurement procedure. All these approaches are based on an additional assumption, namely that the repetition error is normally distributed. One approach is again to compute a 95% range. Another approach is to consider the distribution of the absolute deviation from the true value and to report characteristics such as the mean, the median, or other percentiles. Percentiles of this distribution are, however, directly related to the percentiles of the measurement error distribution itself—for example, the upper 5% percentile of the absolute deviation is identical to the upper bound of the 95% range. The left side of Table 1 summarizes the rules to compute some of these numbers.

Further approaches are based on the idea to consider the (theoretical) distribution of the difference between two replicates. This distribution may be easier to interpret for a user of a measurement procedure than the repetition error distribution itself. The latter refers to the abstract concept of a true mean, whereas the first refers to a distribution with a real counterpart, namely the differences we would observe if we replicate the measurement procedure in practice. Moreover, the distribution of these differences is completely determined by the distribution of the measurement error; in particular, the standard deviation of the differences is just given by $\sqrt{2}\sigma_{\epsilon }$. Consequently, the characteristics we could compute with respect to the distribution of the repetition error can be computed in an analogous manner for the distribution of the differences, as shown on the right hand side of Table 1.

There is also a theoretical advantage in moving from the repetition error to the difference between replicates: differences tend to be closer to a normal distribution than raw measurements. This is illustrated in Figure 2. It can be observed that, even if the repetition error distribution deviates substantially from a normal distribution, the difference between replicates is much closer to a normal distribution, in particular with respect to the 2.5% and 97.5% percentiles.

A final approach to translate $\sigma_{\epsilon }$ to a number easier to interpret is to ask the question of how far two values $\mu_{1}<\mu_{2}$ for two different subjects have to be apart, such that the probability to observe, for the second subject, a larger measured value than for the first subject is 95%. This number is equal to $u_{0.95}\sqrt{2}\sigma_{\epsilon }$, with $u_{0.95}\sqrt{2}=2.327$. This number is often referred to as *Smallest Detectable Difference*, *Minimal Detectable Difference*, or *Minimal Detectable Change*. It should not be confused with the *Minimal Important Difference (MID)* [12,13].

## 9. Relating the Repetition Error to the Population Variation

In Section 4, we have pointed out that it is necessary to relate any quantification of the measurement error to the magnitude the user may accept in the context of the application of the procedure. However, this requires the familiarity of the user with the quantity to be measured by the procedure. Obviously, this cannot be the case if a procedure provides access to a new quantity for the first time. For example, this situation is typical for a new psychometric instrument.

In this case, it is common to take the population variation of the quantity of interest into account and to relate the standard deviation of the repetition error to this variation. The rationale behind this is rather simple: measurement procedures are used to distinguish between subjects with different true values, and the smaller the measurement error relative to the population variation, the better we can discriminate between subjects with different true values.

There are at least two different ways to quantify the population variation. We can consider the population variation of measurements of this quantity, or the variation in the true mean values $\mu_{i}$ (which may coincide with the true values in the optimal case). The latter can be estimated by adding the assumption $\mu_{i}∼N\left(\mu ,\sigma_{p}\right)$ to the simple repetition error model (1). Such a model is then called a random effects model or a variance component model, and the restricted maximum likelihood (REML) method allows us to obtain estimates of both $\sigma_{p}$ and $\sigma_{\epsilon }$. The standard deviation of the population variation of the measured quantity is then given by $\sqrt{{\sigma_{p}}^{2}+{\sigma_{\epsilon }}^{2}}$. Hence, we can relate the standard deviation of the repetition error to the population variation by considering the ratios
$$\xi_{1}:=\frac{\sigma_{\epsilon }}{\sigma_{p}}\;\mathrm{or}\;\xi_{2}:=\frac{\sigma_{\epsilon }}{\sqrt{{\sigma_{p}}^{2}+{\sigma_{\epsilon }}^{2}}}$$

However, the most popular way is given by the so called intraclass correlation coefficient (ICC), defined as
$$\mathrm{ICC}:=\frac{{\sigma_{p}}^{2}}{{\sigma_{\epsilon }}^{2}+{\sigma_{p}}^{2}}$$
which is equal to $1-{\left(\xi_{2}\right)}^{2}$. Consequently, the ICC behaves differently compared to the numbers shown in Table 1 and the ratio $\xi_{1}$. Those numbers are all proportional to $\sigma_{\epsilon }$, i.e., if the measurement error increases by a factor of two, these numbers are all also doubled. In contrast, the ICC decreases with increasing measurement error, and the relation is non-linear.

However, there is a slight conceptual issue with the idea of taking the population variation into account. $\sigma_{p}$ obviously depends on the population chosen for the study. We argued above that the study population of a measurement error study should represent a certain relevant subject population, and if this advice is followed, we also have a clear interpretation of $\sigma_{p}$. However, there may be subtle differences. In designing a measurement error study, we aim to be representative with respect to the magnitude of measurement error. This may imply that we do not care a lot about subject characteristics we believe are unrelated to this magnitude. For example, if we consider a procedure to measure a physiological parameter in clinical populations, it might be acceptable to use patients from one single hospital. However, the distribution of the physiological parameter may depend on whether the hospital is located in a rural or an urban area.

In any case, using the population variation $\sigma_{p}$ in order to facilitate the interpretation of the repetition error increases the need for a careful and responsible choice of the study population: Increasing the heterogeneity of the population makes the relation between $\sigma_{\epsilon }$ and $\sigma_{p}$ to look more favorable.

## 10. Analyzing Pairwise Differences

In view of the advantages of differences between replicates mentioned above, it might be argued that we should avoid any analysis of the raw measurements and directly use the differences between replicates. Indeed, this is a very popular approach in studies with only two replicates, and we will come back to this specific situation later in Section 14.

However, this approach is also applicable in the case of more than two replicates. We can just build all pairwise differences between replicates, i.e., the differences
$$\delta_{ir_{1}r_{2}}=y_{ir_{2}}-y_{ir_{1}}$$
over all subjects *i* and all pairs $\left(r_{1},r_{2}\right)$ of replicates with $r_{1}\neq r_{2}$. (Note that any pair will enter twice, with opposite roles of the two replicates.) Indeed, the empirical distribution observed in these values is a perfect approximation for the true theoretic distribution of the differences. In particular, it allows us to determine the percentiles of this distribution without any normality assumption in a rather efficient manner. It can, hence, present a valuable alternative—at least for this specific purpose.

## 11. Informing about the Shape of the Repetition Error Distribution

When considering the case of having access to a gold standard method in Section 3, we pointed out that information about the shape of the measurement error distribution can be essential for the user. For example, a skewed error distribution will tell the user about the risk to observe large deviations in particular in one direction. The inspection of the distribution of residuals from a regression model is a well known general statistical technique to study the shape of the error distribution, but this technique has to be handled with care when considering the simple repetition error model (1). Since we estimate, for each subject, its own mean value and the residuals are defined as the differences from this mean value, the distribution of the residuals does not approximate the error distribution, even in large samples. This does only hold if we have a sufficient number of replicates. In particular if there are only two replicates, the distribution of the residuals is always symmetric and we cannot expect to detect a skewed repetition error distribution. However, as soon as there are three or more replicates, we can detect essential characteristics of the shape in the distribution of the residuals, as illustrated in Figure 3.

## 12. Learning about the Repetition Error by Modeling

The simple repetition error model (1) is the perfect basis for estimating the standard deviation of the repetition error. However, it can be extended in several directions, allowing us to get more insights into the behavior of the repetition error.

We already discussed, in Section 2, that the magnitude of the measurement error may depend on patient characteristics. Hence it is a straightforward idea to allow $\sigma_{\epsilon }$ to depend on subject characteristics. For example, if we are interested in investigating the relation of the magnitude of the repetition error to gender, we may fit the model separately for males and females. More generally, we can allow the repetition error $\sigma_{\epsilon }$ to vary from subject to subject, i.e., we replace $\sigma_{\epsilon }$ with ${\sigma_{\epsilon }}_{i}$ in (1) and try to link these individual values to subject-specific characteristics. For example, if we are interested in the relation of the magnitude of the repetition error to age and gender, we can formulate a corresponding linear model for the standard deviation of the log scale, i.e.,
$$\log{\sigma_{\epsilon }}_{i}=\beta_{0}+\beta_{\mathrm{age}}{\mathrm{age}}_{i}+\beta_{\mathrm{gender}}{\mathrm{gender}}_{i}.$$

This way, it becomes feasible to give more specific information to the users of the measurement procedure.

We also discussed, in Section 2, that the magnitude of the measurement error may depend on the observer. There may be systematic differences between observers (i.e., some observers tend to produce, on average, higher values than others) as well as differences in the magnitude of the standard deviation. This suggests considering models in which each observer has her or his own mean $\alpha_{r}$ and own standard deviation ${\sigma_{\epsilon }}_{r}$ of the repetition error. Such a model may read
$$y_{ir}=\mu_{i}+\alpha_{r}+\epsilon_{ir}\;\mathrm{with}\;E\;\epsilon_{ir}=0\;\mathrm{and}\mathrm{Var}\;\epsilon_{ir}={\sigma_{\epsilon }}_{r}^{2}.$$

The inspection of the estimates of $\alpha_{r}$ and ${\sigma_{\epsilon }}_{r}$ may inform us about the single observers *r*, but typically it is of interest to relate these values to specific characteristics of the observers. This can be again approached by formulating corresponding linear models for $\alpha_{r}$ or $\log{\sigma_{\epsilon }}_{r}$. This information may be also relevant for the user of the measurement procedure, if the choice of the observer can be influenced in a daily routine—for example, by ensuring sufficient experience.

Learning about the influence of characteristics of the observers this way requires a sufficient variation in observer characteristics and hence a rather substantial number of different observers. With a limited number of observers, we can still ask the more simple question of whether there are any differences between the observers. This requires only modeling $\alpha_{r}$ or ${\sigma_{\epsilon }}_{r}$ as random effects. In particular, the first choice allows us to decompose the repetition error into two components: the variation in mean values between the observers and the additional variation that cannot be explained this way. If there is substantial variation in mean values, we can hope to reduce the repetition error in the long run by teaching observers to reduce these differences, i.e., we have to convince them to calibrate each other.

The same approach can be used in the case of repetitions over time: $\alpha_{r}$ and $\sigma_{r}$ can be modeled as a function of time. Establishing an effect of time indicates a deviation from perfect replicates. If, in addition, there are also different observers, even the interaction between observer and time can be investigated, i.e., to which degree different observers change differently over time.

A final issue we can investigate is a potential relation between the magnitude of the measurement themselves and the standard deviation of the repetition error. Such a relation is often present, but the degree is often unclear. In particular it would be of interest whether the standard deviation of the repetition error $\sigma_{i}$ is approximately proportional to $\mu_{i}$. In this case, it would be useful to inform the user about the proportionality factor instead of an overall standard deviation. This question can be approached by modeling $\log{\sigma_{\epsilon }}_{i}$ as a linear function of $\mu_{i}$ and to study the intercept and the slope.

Standard software for mixed models (such as PROC MIXED in SAS or the mixed command in Stata) allows us to fit many of these models—as long as the standard deviation is modeled only as a function of one categorical variable. More complex modeling of the standard deviation requires us to use specific software, such as the R-package gamlss [14] or use flexible programs for Bayesian model fitting.

It should be noted that the use of any of these models does not imply that model (1) is incorrect. Model (1) still provides an estimate of the standard deviation of the repetition error for a randomly chosen subject and a randomly chosen observer or time point. All the other models answer questions about the measurement error of a subject, observer or timepoint with a specific characteristic.

## 13. Splitting Up the Measurement Error

Some measurement procedures consist of several steps, which can be replicated separately. For example, in taking an image, the patient has to be positioned correctly, an image has to be taken, the image has to be presented to an observer, the observer has to mark a region of interest, and finally an algorithm computes a number. Each step can be replicated, i.e., the patient can be positioned twice, the image can be taken twice, a randomly chosen set of observers is looking independently at the images, the observers are marking the region of interest twice, and the algorithm is applied twice. Each step has then its own repetition error, and this can be estimated by extending the simple repetition error model (1), in a way writing the error term as a sum of step-specific error terms (variance components). Again, standard statistical software for fitting random effect models can be used to obtain estimates for the standard deviation of the repetition error in each step. It is essential that it is not necessary to replicate each step in each subject. It is sufficient to repeat in each subject one or a few (randomly or pre-specified) selected steps, such that, at the end, there is a sufficient overall number of replicates for each step to reliably estimate the relevant variance components.

## 14. Studies with Two Observers

The vast majority of scientific investigations about the measurement error are based on only two replicates. This is not an issue when we use replicates in time, but it is an issue if we select only two observers. If we select two observers, it is rather unlikely that we can regard this as a random sample form a relevant population of observers. A typical set-up you can find in many investigations of new measurement procedures is to choose as observers the inventor of this new procedure and his or her closest collaborator. Obviously, if in such an investigation a low repetition error is observed, this will not inform potential users about the repetition error they can expect in the future in daily routine. They provide another type of information, which can be useful, too—for example, if we investigate whether a clinician and a radiologist can come to the same judgment based on a radiological image.

Studies with “replicates” based on two observers can be seen as a special case of method comparison studies. In method comparison studies, two different measurement procedures are applied in each subject, and the aim is to inform potential users about the difference they have to expect between the two procedures. There is little danger that a method comparison study is interpreted as an attempt to investigate the measurement error of each procedure. Hence, by regarding the two observers as two different methods, it becomes more obvious that studies with two observers do not aim at informing about the measurement error of the measurement procedure used.

Comparisons of two observers—or method comparison studies—require a different analytical approach compared to studies using replicates: it makes now sense to investigate whether there is a systematic difference between the two observers or methods, respectively. Consequently, the statistical repertoire now includes methods such as a t-test-based confidence interval for the difference in mean values. The very popular approach of the *Limits of Agreement*, introduced by Bland and Altman [15] is just a generalization of the 95%-range mentioned on the right side of Table 1, taking the systematic difference into account. In general, the methodology for analyzing such studies is both similar but also different to studies using replicates. It is beyond the scope of this paper to go into detail here.

Additionally, if more than two methods or more than two purposeful selected observers are compared, we do not approach investigations based on replicates. These studies should be seen as pairwise comparisons. However, it can be very useful to add replicates in the design of a method comparison study, as then it becomes possible to study the repetition error of each measurement procedure and to relate the observed differences between the methods to the repetition error. In particular, it allows us to judge whether the differences can be purely explained by the repetition error. Bland and Altman [15] presented this proposal in 1986, but regretfully, by 2003, their proposal had not been adopted widely [3].

## 15. Using Replicates in Studies with a Gold Standard Measurement

Even if a gold standard measurement procedure is available, it can be of interest to generate in addition replicates for the measurement procedure of interest, following some standard principles, such as using several observers or several time points. First, we can now also estimate the characteristics of the repetition error, allowing comparisons with other investigations reporting such values. Second, we can check whether the principle is able to generate perfect replicates. However, imperfect replicates are often not due to an imperfect way of generating replicates, but due to the (undue) influence of subject characteristics on the measurement procedure. We discussed, in Section 5, potential correlations of the measurement error with factors such as age, gender or education, which we will never detect using only replicates. If a gold standard measurement is available, we can detect this by separately analyzing the bias in males and females. More generally, we can relate the individual measurement error $e_{i}$ to subject characteristics, similar to the approaches outlined in Section 12. However, this obviously does not require the use of replicates. The advantage of using replicates is that we can now split up the individual measurement error $e_{i}=y_{i}-y_{i}^{*}$ into the two components $y_{i}-\mu_{i}$ and $\mu_{i}-y_{i}^{*}$, i.e., into the repetition error and a “systematic” component, and only the latter is affected by subject characteristics. Actuallly, using replicates allows us to estimate this systematic component with much higher precision, compared to the situation of a single measurement in each subject. Hence, we gain substantial power in investigating the relation to subject characteristics, in particular if the spread of the repetition error is large compared to the inter-individual variation in the systematic component. Thus, replicates are used here to decrease the stochastic uncertainty.

## 16. Discussion

### 16.1. How Well Do We Inform Potential Users?

Our considerations suggest that it is relatively simple to inform potential users of a measurement procedure about the measurement error using replicates. Always having in mind that the spread of the repetition error distribution tends to underestimate the spread of the measurement error distribution, information on the standard deviation of the repetition error can be provided and it can be transformed into quantities supporting the interpretation of the magnitude of the repetition error in the context of the application of the procedure by the user.

If it is so simple, why was it necessary to write this paper? One simple answer is that the simple repetition error model (1) is rarely presented in papers discussing the methodology of scientific investigations of the measurement error—although it is, of course, always present in the background. This may make it difficult to explain quantities derived from this model in a simple manner. There are many historical and conceptual reasons for this. For example, for decades, the ICC played a very prominent role in the analysis of measurement error studies [16,17,18], and hence many papers take this as a starting point—in our consideration the ICC is the measure most apart from the standard deviation. On the other hand, the Standard Error of Measurement is often only presented as one of many measures that are computable, masking its fundamental conceptual role as the origin of all other quantities. Moreover, the focus is often on how to compute one measure from each other—reflecting a practical need—which gives a natural focus on technicalities.

There are, of course, also conceptual issues, which we have also touched upon. The thinking in differences between replicates is rather attractive from a pedagogical point of view, as the measurement or repetition error is a rather abstract quantity, involving the concept of true values or the expected value of replicates, respectively. Even from a statistical point of view, an analysis based on differences can have advantages. If the intended use of a measurement procedure is the assessment of a change over time (e.g., after an intervention), then a potential user would probably expect to be informed by a difference approach using a study based on replicates mimicking the relevant time frame. Indeed, in such a situation, considering the measurement error from a study based on a gold standard measurement can be misleading.

In the following sections, we comment in more detail on some of the issues involved in explaining the discrepancy between the existence of a rather clear and simple framework as outlined in this paper and the somewhat confusing situation with respect to explaining and reporting the methodology of scientific investigations on the measurement error of a procedure.

### 16.2. A First Step towards Terminology

We mentioned, in the introduction, the confusion about terminology with respect to scientific investigations of the measurement error. Can the considerations presented in this paper be helpful in this context? Roughly speaking, terminology should assist inter-human communication by ensuring that the same things have the same name and different things have different names. So, a first step should be to reflect about which distinctions are relevant. Based on the considerations presented in this paper, we identified four issues where a distinction might be useful and relevant. Table 2 presents the corresponding topics, the categories to be distinguished, and the reasons why we regard these distinctions as relevant—they have distinctly different consequences.

### 16.3. The Relation to Existing Terminology

Having identified four relevant distinctions, it is natural to ask whether existing terminology is (already) related to these distinctions. We are not aware of terms reflecting the imaginability of true values. However, this distinction coincides roughly with two different user domains, namely those who measure physiological or other “real” parameters and those using psychometric instruments and/or patient reported outcomes and experience measures. With respect to study designs, we can imagine that many people using the terms “reproducibility study” or “repeatability study” actual refer to studies based on using replicates, whereas studies involving a gold standard measurement procedure may be typically called “accuracy study” or “measurement error study”. Since the type of error analyzed matches nearly 1:1 with the study design, these terms may be also interpreted in relation to this type. In scientific fields with imaginable true values, the term “precision” often refers to the repetition error, and “accuracy” to the measurement error.

The distinction between the two possibilities to report characteristics of the repetition error has attracted some attention in the literature. It is mainly reflected in the distinction between “agreement measures” and “reliability measures” [7,19]. We are not very happy about the use of the term “agreement” to characterize the reporting of the characteristics of the repetition error. The term “agreement” typically refers, in common language, to a specific relation between two people or two parties. So, when talking about “agreement”, it is rather natural to think in differences, and hence the term contributes to the lack of popularity in thinking in terms of the repetition error. In our opinion, it would be useful to use the term “agreement” only with respect to studies with the purposeful selection of single observers or measurement procedures, as here the task is indeed to investigate pairwise agreement. This applies, in particular, if the term “agreement” is not just used in connection with the term “measure”, but in connection with the term “study”. The contrast to “agreement studies” may then be characterized by the terms “reproducibility study” or “repeatability study”, emphasizing the intention to generate perfect replicates.

Whether the term “reliability measure” is adequate or not to characterize measures, also depending on the population variation, cannot be judged based on our considerations. We touch on measures, such as the ICC, only from one specific perspective, namely how to inform users about the repetition error in the case of lacking familiarity with the quantity of interest. In the literature, the relation between the repetition error and the population variation is often seen as a scientific question of its own interest [20], related to the distinction between the evaluative and the discriminative use of measurement procedures [21]. To which degree these distinctions and terms are useful cannot be addressed within the scope of this paper.

As the terms “agreement” and “reliability” may suggest unintended associations, it might be worth thinking about alternative terms to distinguish these two types of reporting characteristics of the repetition error. One idea would be the terms “absolute measures” and “relative measures”. This would also reflect that the absolute measures have a unit identical to that of the measurements themselves, whereas the relative measures are unit-free ratios. Indeed, these terms can be also found in the literature [1,20]. In any case, it should always be emphasized that such terms reflect a distinction, but they do not exclude each other. Any scientific investigation of the measurement error based on replicates allows for estimating the magnitude of the repetition error. Even if relative measures are of primary interest in a specific situation, absolute measures can and should also be reported.

It is not the intention of this paper to introduce a new term for the type of studies described in this paper. The paper just presents a specific view on a range of existing studies (or study types), focusing on the common aim to inform potential users about the measurement error of a single measurement procedure by trying to replace an absent gold standard method by some type of replicates. This allows for highlighting the similarity among all studies following this principle with respect to the major conceptual approach, as well as the statistical analysis and may contribute to a better understanding of some common issues. It is, of course, always somewhat arbitrary to define the scope of such a view. For example, we decided to consider both replicates in time as well as replicates using different observers in our view. This requires a somewhat abstract view, but we felt that, in our context, the similarities are larger than the dissimilarities. (Although, in Section 12, one difference becomes obvious: repetitions in time have time as an underlying continuous scale, which is not present when using observers.) This does not mean that we should not distinguish between these two settings in general. For example, “repeatability” may be used to characterize repetitions under similar conditions (i.e., over time), and “reproducibility” to characterize repetitions under varying conditions (i.e., observers), cf. [7]. In general, it should be noted that there is no general consensus about which general aspect of a study should be reflected by terms for study types. Terms such as “case-control study”, “cohort study” or “randomized trial” reflect basic designs, and terms such as “risk factor studies” or “diagnostic accuracy study” reflect a specific aim, whereas terms such as “treatment study” or “epidemiological study” may reflect rather general aims. The confusion about terminology in the field of measurement error studies is, in our opinion, partially due to the fact that it is often unclear whether terms such as “agreement study”, “reliability study” or “reproducibility study” refer to designs or to aims.

### 16.4. The Move from Two Replicates to Many Replicates

Studies comparing two observers (or two measurement procedures) have played a dominant role in the history of measurement error studies, and the statistical approaches and methods discussed in the literature are hence mainly reflecting this situation. Fortunately, it has been recognized, to an increasing degree, that inter-observer reliability studies have to be based not only on two but several observers in order to allow a generalization in a meaningful manner and to inform potential users. However, the analytical thinking used in presenting results from these studies often still reflects the idea of looking at differences, e.g., by transforming the standard deviation of the repetition error into quantities related to the difference between replicates or between two subjects. The simple repetition error model is rarely mentioned explicitly, and, in this way, researchers and readers are not aware of the additional possibilities offered by such an approach with respect to getting deeper insights into the repetition error, as described in Section 11, Section 12 and Section 13. We hope and expect that the increasing use of many observers contributes to making the thinking in the simple repetition error model more popular.

### 16.5. Using the Simple Repetition Error or the Difference Approach?

This may also contribute to the confusion that there are, indeed, two different statistical approaches to the analysis of scientific investigations using replicates, amongst which we cannot say that one prevails over the other. We acknowledge that the difference approach in the form presented in this paper may not have been widely used for analyzing studies with more than two replicates. Moreover, for studies with two replicates, the difference approach is used, typically without the step of considering all pairs in both orders.

We did not present a systematic comparison of the two approaches in this paper. However, we would like to mention two further properties of the fully pairwise difference approach. First, there are no standard technical techniques to perform statistical inference, i.e., to compute confidence intervals. Second, some of the modeling approaches mentioned in Section 12 can also be easily applied when using the differences, whereas some are more challenging. The latter happens in particular if there is a need to model the effect of covariates that are not constant within pairs.

### 16.6. Neglected Topics

We could not touch all topics that may be relevant in the context of using replicates to inform potential users of a measurement procedure. In particular, we focused on the definition of key quantities and their estimation, but did not address the question of communicating uncertainty of parameter estimates. Indeed, this is partially ignored in reporting the results of scientific investigations of measurement error studies, i.e., estimates are presented without standard errors or confidence intervals. This is partially due to the statistical software used: Programs for ANOVA or regression typically do not report such values for the RMSE. From this point of view, it is preferable to use software for random effect models, often providing this information. The statistical validity of estimates and inference procedures has recently attracted some attention with respect to the analysis of method comparison studies [22,23,24]. It is desirable that these investigations are extended to the estimates based on the simple repetition error model mentioned in this paper. The same holds for sample size considerations [25,26].

We also only considered the case of measurement procedures, resulting in continuous numbers. Many measurement procedures finally result in a binary (or categorical) decision, and the error at this level is of high interest for the user. The considerations of this paper cannot be transferred directly to the binary case. The average of binary replicates provides as estimate of a probability, and such a probability cannot be regarded as a substitute for a true binary value. Moreover, the repetition error is a difference between a binary variable and a probability, such that the standard deviation may be not a natural way to characterize this distribution. Hence, alternative perspectives are necessary to explain the value of replicates for a binary measurement procedure—for example, the formal relation between the ICC and the kappa statistic [27,28].

## Figures and Tables

**Figure 1 diagnostics-11-00162-f001:**
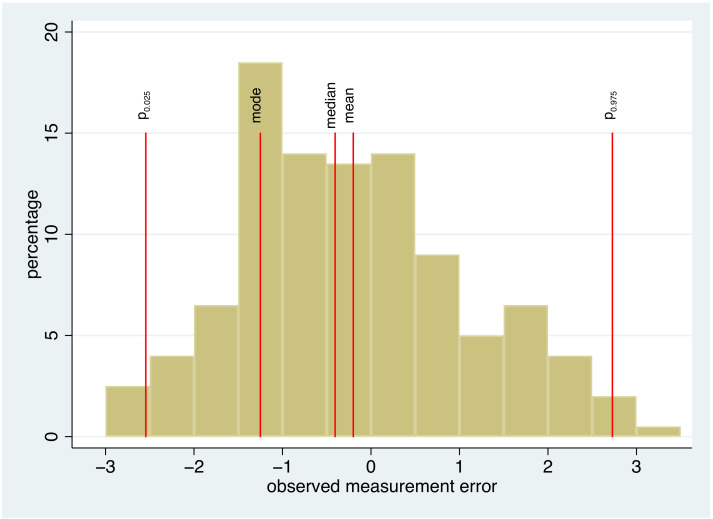
A histogram of the observed distribution of the measurement error in a study with gold standard measurements. The 2.5% percentile, the mode, the median, the mean, and the 97.5% percentile are marked by red lines.

**Figure 2 diagnostics-11-00162-f002:**
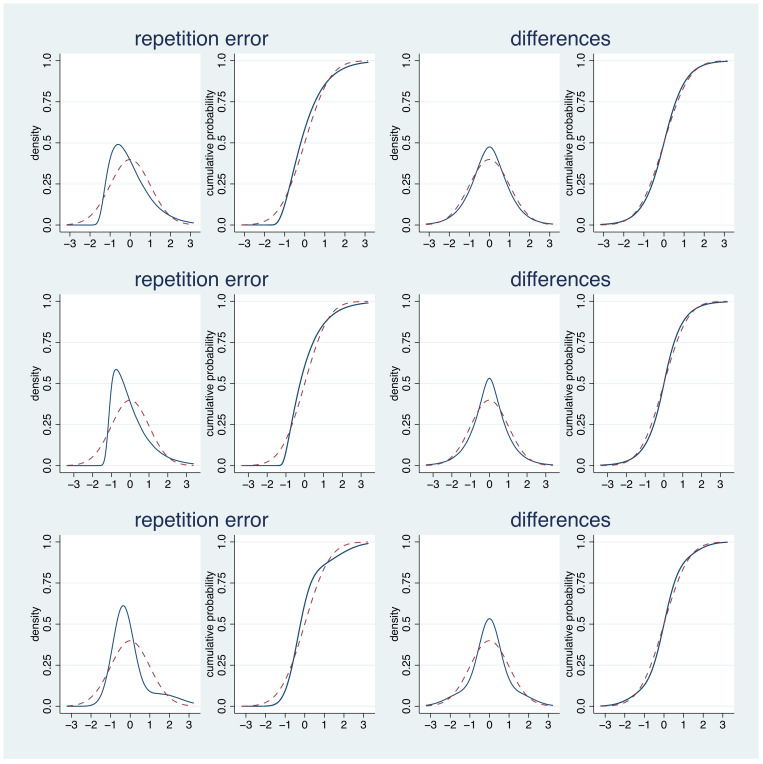
The density and cumulative distribution functions of three non-normal repetition error distributions and the corresponding functions for the distribution of the difference between two replicates. All distributions are standardized to a standard deviation of 1. The red dashed lines indicate the corresponding functions for a standard normal distribution as assumed when transforming the estimated standard deviations.

**Figure 3 diagnostics-11-00162-f003:**
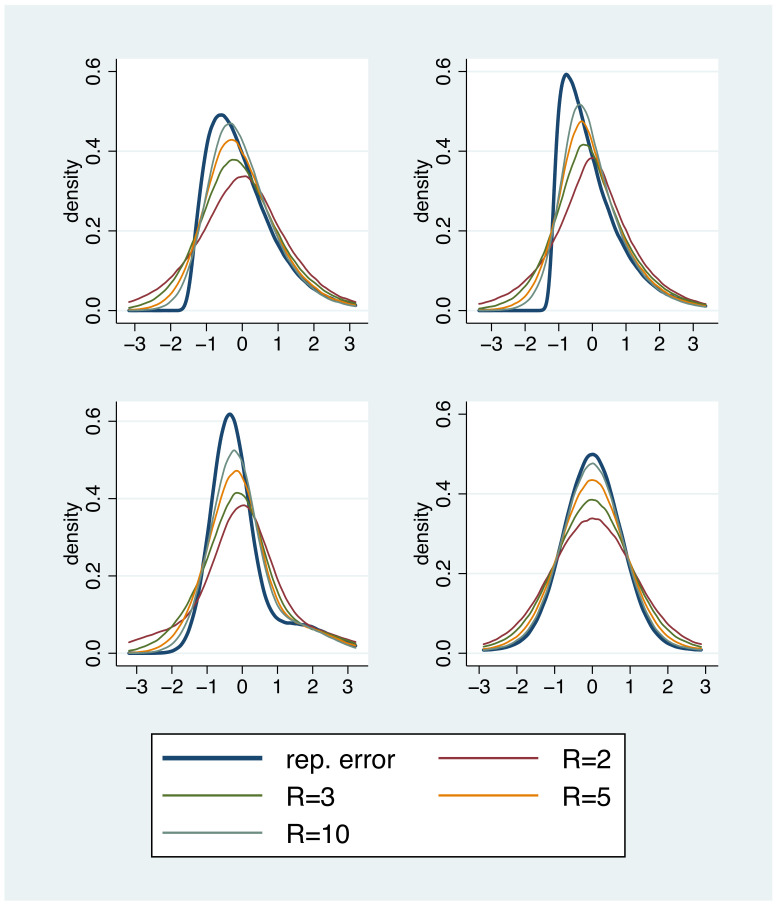
Four repetition error distributions and the corresponding distributions of residuals based on *R* replicates. Residuals were rescaled by multiplication with the factor $\frac{R}{R-1}$.

**Table 1 diagnostics-11-00162-t001:** Distributional characteristics of the repetition error and the difference between replicates expressed as a function of the standard deviation $\sigma_{\epsilon }$ of the repetition error. $u_{\alpha }$ denotes the α-quantile of a standard normal distribution. For concrete computations, the following approximate values can be used: $\sqrt{2}=1.4142$, $u_{0.025}=-1.96$, $u_{0.975}=1.96$, $u_{0.025}\sqrt{2}=-2.7718$, $u_{0.975}\sqrt{2}=2.7718$, $\sqrt{\frac{2}{\pi }}=0.7978$, $\frac{2}{\sqrt{\pi }}=1.1284$, $u_{0.75}=0.6745$, $u_{0.75}\sqrt{2}=0.9539$. *: When the standard deviation of the repetition error is reported, often the term *Standard Error of Measurement (SEM)* is used. **: The half width of this interval is often reported as the repeatability coefficient when using two replicates over time [11].

	Variable Considered	
**Distributional Characteristic**	**Repetition Error**	**Difference between Replicates**
standard deviation	$\sigma_{\epsilon }$ *	$\sqrt{2}\sigma_{\epsilon }$
95%-range	$\left[u_{0.025}\sigma_{\epsilon },u_{0.975}\sigma_{\epsilon }\right]$	$\left[u_{0.025}\sqrt{2}\sigma_{\epsilon },u_{0.975}\sqrt{2}\sigma_{\epsilon }\right]$ **
mean absolute value	$\sqrt{\frac{2}{\pi }}\sigma_{\epsilon }$	$\frac{2}{\sqrt{\pi }}\sigma_{\epsilon }$
median absolute value	$u_{0.75}\sigma_{\epsilon }$	$u_{0.75}\sqrt{2}\sigma_{\epsilon }$

**Table 2 diagnostics-11-00162-t002:** Four relevant distinctions in communication about systematic investigations of the measurement error of a measurement procedure.

Topic	Categories	Relevant Consequences
existence of true values	true values are imaginable	– question about existence of gold standard measurement procedure is reasonable – concept of “perfect replicates” is clearly defined
	true values are not imaginable	– no reason to look for gold standard method – distinction between measurement error and repetition error conceptually challenging
study design	gold standard measurement procedure applied	– measurement error can be analysed
	replicates used	– only repetition error can be analysed, which can be smaller than the measurement error
	both	– both can be analysed and relation to subject characteristics can be studied with higher power
type of error analysed	measurement error	– systematic bias can be analysed – numbers reported can take systematic bias into account
	repetition error	– systematic bias cannot be analysed – numbers reported all assume a systematic bias of 0 – depends on choice of the manner to generate replicates
reporting characteristics of repetition error	only depending on repetition error	choice of measures: SEM or transformations
	also depending on population variation	choice of measures: ICC or related measures

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
