# Peer review of "How Replicates Can Inform Potential Users of a Measurement Procedure about Measurement Error: Basic Concepts and Methods"

_diagnostics, 2021, doi:10.3390/diagnostics11020162_

Round 1

Reviewer 1 Report

The Authors have described the concept, use and communication of potential measurement errors.

Main finding:

The authors should somehow try to define the term "gold standard" with examples, if possible from e.g. an SI viewpoint as in a size of a tumor, a gold standard would be a 3D measueremnt in m3?

Or e.g. in blood pressure measurement N/m2 is quite straightforward. Or maybe the concept of gold standard is not clear.

A brief chapter would probably help in the understanding.

Minor findings:

Some grammar or rephrasing mistakes need to be corrected in:

Line 123

Line 188

Line 194

Line 210

Line 258

Line 271 and 281

Line 336

Line 395

Author Response

We are very grateful to the reviewer for the careful reading of the manuscript and the constructive comments. We agreed with most of the suggestions of the reviewer.

Plese find our point-to-point response in the pdf-file attached.

Reviewer 2 Report

Please refer to the attached manuscript with comments marked in.

Author Response

We are grateful to the reviewer for the careful reading of the manuscript. Please find below our point-by-point response.

Main finding:

The authors should somehow try to define the term "gold standard" with examples, if possible from e.g. an SI viewpoint as in a size of a tumor, a gold standard would be a 3D measueremnt in m3?

Or e.g. in blood pressure measurement N/m2 is quite straightforward. Or maybe the concept of gold standard is not clear.

A brief chapter would probably help in the understanding.

We agree with the reviewer that we have been a little bit fuzzy with using the term „gold standard“. We now use consequently the term „gold standard measurement procedure“ (and sometimes for short the term „gold standard method)“ to emphasize that we are talking about a measurement procedure. We refer also to the values produced by this method as „gold standard measurements“.

We also explain this term more precisely at it first appearance in the main text:

If a gold standard measurement procedure is available, i.e. a method allowing us to measure the true value of the quantity of interest in an error-free manner, ...

We did not include a separate chapter on this topic, as issues related to finding gold standard methods are already discussed in Section 6.

Minor findings:

Some grammar or rephrasing mistakes need to be corrected in:

Line 123

„recommend" was changed to „recommended"

Line 188

„an pragmatic” was changed to „a pragmatic"

Line 194                  

„with other words" was changed to „in other words"

Line 210

„we" was deleted

Line 258

The spelling of „repititiion" was corrected

Line 271 and 281

„increase" was changed to „increases”; „my" was changed to „may"

Line 336

„im" was changed to „in"

Line 395

„investigating“ was changed to „investigation"